# Acute Malnutrition in Under-Five Children in KwaZulu-Natal, South Africa: Risk Factors and Implications for Dietary Quality

**DOI:** 10.3390/nu17122038

**Published:** 2025-06-18

**Authors:** Meshack Mzamani Mathosi, Lindiwe Priscilla Cele, Mmampedi Mathibe, Perpetua Modjadji

**Affiliations:** 1Department of Public Health, School of Health Care Sciences, Sefako Makgatho Health Sciences University, 1 Molotlegi Street, Ga-Rankuwa 0208, South Africa; 2Non-Communicable Diseases Research Unit, South African Medical Research Council, Tygerberg, Cape Town 7505, South Africa; 3Department of Life and Consumer Sciences, College of Agriculture and Environmental Sciences, University of South Africa, Florida Campus, Johannesburg 1709, South Africa

**Keywords:** acute malnutrition, dietary diversity, Infant and Young Child Feeding, socioeconomic status, under-five children, primary health facilities, South Africa

## Abstract

Background/Objectives: Despite ongoing national interventions, pockets of acute malnutrition persist in rural settings of South Africa, contributing to disproportionate rates of child morbidity and mortality. This study aimed to identify risk factors associated with acute malnutrition among under-five children attending primary healthcare facilities in Msinga, KwaZulu-Natal Province, with a specific focus on dietary diversity and selected infant and young child feeding indicators. Methods: A cross-sectional, facility-based study was conducted among 415 mother–child pairs attending five randomly selected PHC facilities in the Msinga sub-district. Participants were selected using a multistage sampling design from a sampling frame of 18,797 under-five children. Of the 551 mother–child pairs approached; the final analytic sample comprised 415 observations. Data were collected through structured interviews, anthropometric assessments, and dietary diversity scores (DDS). Data were analyzed using Stata 18, and Poisson regression was applied to identify risk factors. Results: The prevalence of acute malnutrition was 29% based on weight-for-height/length z-scores (WHZ/WLZ) and 27% based on mid–upper-arm circumference z-scores (MUACZ). Children aged ≥36 months had significantly higher prevalence of acute malnutrition (aPR = 1.62; 95% CI: 1.15–2.10). Children from households with five or more members had reduced risk (aPR = 0.66; 95% CI: 0.45–0.74), and those born full-term showed a strong protective association (aPR = 0.39; 95% CI: 0.23–0.64). Maternal age was associated with reduced risk, with children of mothers aged 25–34 years (aPR = 0.67; 95% CI: 0.48–0.93) and ≥35 years (aPR = 0.58; 95% CI: 0.35–0.84) experiencing significantly lower prevalence. Mixed feeding was significantly associated with reduced risk (aPR = 0.86; 95% CI: 0.55–1.17), while a DDS ≥4 was protective (aPR = 0.41; 95% CI: 0.04–0.84). Consumption of protein-rich foods was notably low, with only 21% consuming flesh foods and 10% consuming eggs. Conclusions: Acute malnutrition in under-five children was significantly associated with poor diet quality, older age, low household income, and mixed feeding practices. Protective factors included full-term pregnancy, larger households, older maternal age, and adequate dietary diversity, highlighting the importance of targeted, multisectoral interventions.

## 1. Introduction

Acute malnutrition remains a critical public health challenge in low- and middle-income countries (LMICs), disproportionately affecting under-five children [1,2]. Clinically, acute malnutrition is characterized by rapid weight loss or failure to gain weight and is classified into moderate acute malnutrition (MAM) and severe acute malnutrition (SAM). Both conditions are strongly associated with increased susceptibility to infections, impaired immune function, poor dietary quality, and elevated risk of mortality [3,4]. Globally, an estimated 45 million children are wasted, including 13.6 million with SAM, with the highest burden concentrated in sub-Saharan Africa (SSA) and South Asia [5]. Children with SAM face a tenfold increased risk of death compared to their well-nourished peers, while those with MAM are approximately three times more likely to die from common infectious diseases [6]. These early-life nutritional insults are also consistent with the Developmental Origins of Health and Disease hypothesis, which posits that adverse exposures during critical periods of growth, such as undernutrition in early childhood, can have lasting effects on metabolic health, cognitive development, and disease risk across the life course [7,8].

In SSA, the persistence of acute malnutrition is driven by a complex interplay of structural and environmental determinants, including entrenched poverty, recurrent food insecurity, climate shocks, and fragile health systems [9,10]. These factors undermine access to diverse, nutrient-rich diets and essential maternal and child health services. Additionally, poor water, sanitation, and hygiene (WASH) infrastructure perpetuates cycles of undernutrition and infection, further compromising child health outcomes [11,12]. The COVID-19 pandemic and ongoing economic instability have exacerbated these vulnerabilities, disrupting nutrition services and placing millions of children at heightened risk of growth faltering and preventable mortality [13].

Acute malnutrition remains a major contributor to under-five mortality in South Africa, particularly in rural provinces such as the KwaZulu-Natal Province. Between 2017 and 2018, severe acute malnutrition (SAM) was implicated in 30.9% of child deaths [14]. South Africa has adopted WHO guidelines for the management of SAM, including ready-to-use therapeutic foods (RUTFs) and community-based management (CMAM); however, implementation remains uneven across provinces [15]. Therapeutic nutrition interventions, such as fortified maize blends and energy-dense supplements, have been introduced at the primary health care (PHC) level, yet their reach and effectiveness in rural settings remain limited [16]. While the national case fatality rate for SAM declined from 13.1% in 2011 to 8% in 2017, poor treatment outcomes persist in under-resourced rural facilities, posing a significant barrier to achieving Sustainable Development Goal (SDG) 3.2 by 2030 [17]. The integration of growth monitoring and promotion (GMP) and nutrition services into PHC has not translated into timely identification and referral of malnourished children. Many are referred late, often presenting with complications such as oedema, diarrhoea, and respiratory infections, which significantly elevate the risk of mortality [18,19].

Several national and provincial strategies aimed at improving child nutrition were in place. These included the Child Support Grant (CSG) [20], which provided monthly financial assistance to low-income caregivers, and the Integrated Nutrition Programme (INP), which promoted growth monitoring, nutrition education, and food supplementation at PHC level [15]. However, implementation challenges, such as delayed food deliveries, poor infrastructure, and limited household-level engagement, were reported in rural districts like Msinga [21]. These limitations draw attention to the need for more integrated, community-based strategies that extend beyond institutional settings to address household-level drivers of malnutrition. Given the persistent burden of acute malnutrition in rural South Africa and the limited empirical evidence linking dietary quality and caregiving practices to child nutritional outcomes in these settings, we hypothesized that inadequate dietary diversity and suboptimal IYCF practices are significantly associated with acute malnutrition in under-five children. Therefore, the objective of this study was to identify the key risk factors associated with acute malnutrition among under-five children attending PHC facilities in the Msinga sub-district of KwaZulu-Natal Province, South Africa. Specifically, this study aimed to examine the associations between acute malnutrition and child-level characteristics, maternal factors, and household-level determinants, with a particular focus on the implications for dietary quality and infant and young child feeding (IYCF) indicators [22,23].

## 2. Materials and Methods

### 2.1. Study Design and Conceptual Frameworks

This study employed a cross-sectional analytical design to investigate the risk factors associated with acute malnutrition in under-five children attending primary healthcare (PHC) facilities in the Msinga sub-district of KwaZulu-Natal Province, South Africa. This study was conducted between November 2022 and May 2023. The source population comprised approximately 18,797 under-five children, who accessed PHC services in the district during the 2020/21 reporting period, as recorded in the District Health Information System (DHIS) [24]. Ethical approval for this study was obtained from the Sefako Makgatho Health Sciences University Research and Ethics Committee (SMUREC) (SMUREC/H/139/2022: PG), approved on 2 June 2022, and permission to access PHC facilities was granted by the Department of Health, KwaZulu-Natal Province (NHRD Ref: KZ_202207_033). Informed consent was obtained from mothers prior to data collection, following a clear explanation of this study’s purpose and after addressing the concerns raised. This study adhered to the ethical principles in the Declaration of Helsinki [25].

The research was conceptually informed by the UNICEF and WHO frameworks on the determinants of child malnutrition, which delineate immediate causes (e.g., inadequate dietary intake and disease), underlying factors (e.g., food insecurity, caregiving practices, and access to health services), and structural determinants (e.g., poverty, education, and governance) [22,23,26]. These models guided the selection of variables and the analytical strategy, enabling a comprehensive assessment of nutritional risk factors in a resource-constrained rural context.

### 2.2. Study Setting

This study was conducted in the Msinga sub-district, located within the uMzinyathi district of KwaZulu-Natal Province. Msinga is a predominantly rural area characterized by high poverty rates, limited infrastructure, and constrained access to health and nutrition services. The sub-district comprises one district hospital, one community health centre, 18 PHC clinics, and four mobile units, serving a population distributed across 20 municipal wards. IsiZulu is the dominant local language. According to the District Health Information System (DHIS), the PHC headcount for under-five children in Msinga was approximately 18,797 in the 2020/21 reporting period [24].

### 2.3. Study Population

This study targeted outpatient under-five children attending PHC facilities in the Msinga sub-district of KwaZulu-Natal Province, South Africa, accompanied by their biological mothers. Msinga is a rural area with a high burden of child malnutrition and limited access to health and nutrition services. The study population was drawn from children attending routine child health services, such as growth monitoring and immunization, across five selected PHC facilities. Eligibility criteria required that the child be accompanied by their biological mother, who was at least 18 years old and willing to provide informed consent. Children were excluded if they were acutely ill at the time of data collection, and mothers were excluded if they were under 18, medically unfit, unable to breastfeed due to illness, or declined to participate. This defined population allowed for the collection of relevant data on maternal and child health and nutrition in a high-risk, underserved setting.

### 2.4. Sample Size and Sampling Procedure

The sample size was determined using the Raosoft sample size calculator [27], based on an estimated population of 18,797 under-five children, attending PHC facilities in the Msinga sub-district during the 2020/2021 reporting period [24]. The calculation assumed a 5% margin of error, a 95% confidence level, and a 50% response distribution, resulting in a minimum required sample size of 377 participants. To account for potential non-responses and incomplete data, a 10% buffer was added, yielding a final target sample size of 415. The recruitment process began with the identification of the study population, which consisted of outpatient children under the age of five attending five PHC facilities in the Msinga sub-district, accompanied by their biological mothers. A multistage sampling strategy was employed to ensure a representative sample of the target population. In the first stage, five PHC facilities were randomly selected from the list of eligible clinics in the sub-district. In the second stage, systematic random sampling was used to select mother–child pairs within each facility. Every third mother in the clinic waiting area was approached and invited to participate after completing her consultation.

A total of 551 mother–child pairs were approached across five facilities. Of these, 131 were excluded for not meeting the eligibility criteria. The remaining 420 mother–child pairs met the inclusion criteria and consented to participate, yielding a recruitment rate of 76.2% and a 100% participation rate among eligible individuals. However, five questionnaires were excluded from analysis due to having more than 10% missing data on primary outcomes, resulting in a final analytic sample of 415 participants. The distribution of participants across facilities was as follows: 81 from facility one, 106 from facility two, 79 from facility three, 79 from facility four, and 75 from facility five. Although five cases were excluded due to missing data, the final analytic sample size (*n* = 415) exceeded the anticipated minimum sample size of 377, ensuring representativeness and sufficient statistical power for this study’s objectives. The recruitment process is illustrated below in Figure 1.

### 2.5. Data Collection

This study adhered to the STROBE guidelines to ensure methodological transparency and completeness [28]. To ensure quality throughout this study, the structured questionnaire used to collect data on maternal and child characteristics, feeding practices, and dietary diversity was adapted from previously validated tools in maternal and child nutrition research. Data were collected from November 2022 to May 2023, using a structured, interviewer-administered questionnaire adapted from validated tools used in maternal and child nutrition research [29,30,31], and considered UNICEF and WHO frameworks on the determinants of child malnutrition [22,23,26]. The instrument captured maternal socio-demographics, household characteristics, obstetric history, and child characteristics. Content and face validity were established through expert review by members of the School of Health Care Sciences Research Committee, who evaluated the instrument for relevance, clarity, and alignment with this study’s objectives. To enhance linguistic and cultural appropriateness, the questionnaire was translated from English into IsiZulu by a bilingual translator fluent in both languages. A pilot study involving 15 mother–child pairs was conducted at a PHC facility not included in the main study to assess the clarity, feasibility, and contextual relevance of the instrument. Feedback from the pilot informed minor revisions to improve the wording and layout of the questionnaire. Research assistants fluent in IsiZulu received one week of intensive training on standardized data collection procedures, including interview techniques and anthropometric measurement protocols. During the pilot phase, their performance was assessed to ensure consistency and accuracy in administering the questionnaire and recording measurements. All anthropometric equipment, including digital weighing scales and stadiometers, was calibrated regularly, and each measurement was taken twice to minimize error and enhance precision. The final version of the questionnaire was administered in IsiZulu by trained research assistants under the supervision of the principal investigator. All anthropometric measurements for children were performed by the principal investigator, who is a qualified nurse with clinical experience in child health assessments. The research assistants were responsible for conducting the interviews and administering the structured questionnaires.

#### 2.5.1. Socio-Demographics and Obstetric History of Study Participants

A structured, interviewer-administered questionnaire used to collect data on sociodemographic variables and obstetric history was adapted from previously validated questionnaires in maternal and child nutrition research [29,30,31]. The tool was designed to capture a comprehensive profile of each participant, encompassing maternal socio-demographic variables (including age, marital status, educational attainment, and employment status), household-level indicators (such as income, household head and size, dwelling type, and access to basic utilities such as electricity and water), obstetric history, and key child characteristics (notably age and sex). Under the supervision of the principal investigator, research assistants conducted face-to-face interviews with mothers, administering the questionnaire in the local language, isiZulu.

#### 2.5.2. Complementary Feeding Practices and Dietary Diversity

Complementary feeding practices were assessed using a structured questionnaire adapted from WHO guidelines on IYCF practices [32,33]. During face-to-face interviews, conducted by the research assistants, mothers were asked to recall all foods and beverages consumed by their child in the 24 h preceding the interview. Key IYCF indicators included timing of complementary food introduction, feeding frequency, continued breastfeeding, and types of foods offered. As per WHO guidelines, complementary feeding should begin at six months, when breast milk alone no longer meets an infant’s nutritional needs. Nutrient-rich foods should be introduced in safe, age-appropriate textures and frequencies, 2–3 times daily for infants aged 6–8 months, increasing to 3–4 times for those aged 9–23 months, with 1–2 additional snacks for children over 12 months [34].

Dietary diversity was assessed using the WHO-recommended 24 h recall method, which captures all foods and beverages consumed by the child in the previous 24 h. While this approach is widely used in population-based nutrition surveys due to its simplicity and feasibility, it is subject to certain limitations, including recall bias and day-to-day dietary variation. Compared to a seven-day recall, the 24 h method may not fully capture habitual intake, potentially underestimating or overestimating dietary diversity [34]. Dietary diversity was assessed using the WHO-recommended 24 h recall method, based on maternal recall of the child’s consumption from seven FAO-defined food groups [35]. A dietary diversity score (DDS) of ≥4 was considered adequate, while ≤4 indicated low diversity. Minimum dietary diversity (MDD), minimum meal frequency (MMF), and minimum acceptable diet (MAD) were calculated following WHO guidelines [35]. MDD was defined as the proportion of children who consumed food from at least four out of seven WHO-recommended food groups within the previous 24 h. The seven food groups included grains, legumes, dairy, flesh foods, eggs, vitamin A-rich fruits and vegetables, and other fruits and vegetables [36].

#### 2.5.3. Anthropometric Measurements and Nutritional Indicators of Children

Anthropometric measurements of children were performed by the principal investigator in accordance with standardized WHO procedures [32]. Weight and height measurements in children were conducted using age-appropriate equipment and standardized procedures to ensure accuracy. For children under 2 years, weight was measured using Seca 354 digital baby scale (Seca GmbH & Co. KG, Hamburg, Germany), with the child undressed or in light clothing. Recumbent length was taken using the Seca 210 infantometer (Seca GmbH & Co. KG, Hamburg, Germany), with the child lying flat, legs extended, and measurement recorded to the nearest 0.1 cm. For children aged 2 to 5 years, weight was measured using a digital scale, while the child stood still, barefoot, and in light clothing. Standing height was measured using a portable Seca stadiometer (Seca GmbH & Co. KG, Hamburg, Germany), with the child standing upright, heels together, and head positioned in the Frankfurt plane. All measurements were taken twice by trained personnel and recorded to the nearest 0.1 kg for weight and 0.1 cm for height or length to ensure precision and consistency.

Anthropometric data were entered into WHO Anthro v3.22 to generate sex-specific z-scores for weight-for-height/length (WHZ/WLZ) and mid-upper arm circumference-for-age (MUACZ). Based on WHO standards: WHZ/WLZ ≥ −2SD = normal; WHZ/WLZ < −2SD = acute malnutrition, −3SD ≤ WHZ/WLZ < −2SD = MAM; WHZ/WLZ < −3SD = SAM; WHZ/WLZ > +2SD = overweight/obesity. Similarly, MUACZ ≥ −2SD = normal; MUACZ < −2SD = acute malnutrition, −3 SD ≤ MUACZ < −2 SD = MAM; MUACZ < −3 SD = SAM [37].

#### 2.5.4. Anthropometric Measurements and Nutritional Indicators of Mothers

Anthropometric data collection for the mothers was carried out by the principal investigator, following the standardized guidelines set by the World Health Organization [38]. Weight was measured using a Beurer GS 203 digital scale (Ulm, Germany), with each mother standing upright, barefoot, and wearing light clothing. The scale was placed on a firm, level surface to ensure accuracy. Two weight readings were taken and recorded to the nearest 0.1 kg, with the average used for analysis. Height was measured using a Seca stadiometer (Hamburg, Germany), with mothers standing barefoot, heels together, back straight, and head aligned in the Frankfort horizontal plane. Two height measurements were taken and averaged, recorded to the nearest 0.1 cm. These measurements were used to calculate the Body Mass Index (BMI) by dividing the weight in kilograms by the square of the height in meters (kg/m^2^). BMI categories were defined as follows: underweight (BMI < 18.5), normal (18.5–24.9), overweight (25.0–29.9), and obese (≥30.0) [38].

### 2.6. Statistical Analysis

Statistical analysis was performed using STATA version 18 (StataCorp, TX, USA), following data entry and cleaning in Microsoft Excel. The analytical population consisted of children under five years of age who met the inclusion criteria and had complete data on key variables. During complete case analysis, five questionnaires with more than 10% missing information, particularly on primary outcomes, were excluded, resulting in a final sample size of 415. Anthropometric indices were calculated using WHO Anthro software to generate standardized z-scores, which were compared by sex and age categories using chi-square tests. The Shapiro–Francia test was used to assess the normality of continuous variables. Due to non-normal distributions, non-parametric tests such as the Mann–Whitney U and Kruskal–Wallis (Wilcoxon rank-sum) were used for group comparisons. To identify risk factors for acute malnutrition, univariate regression was first conducted to assess associations between each independent variable and the outcome. Variables with a *p*-value < 0.25 were included in the multivariable model to avoid excluding potential confounders. Adjusted prevalence ratios (aPR) were estimated using a generalized linear model (GLM) with a Poisson distribution and log link function, applying robust standard errors to account for overdispersion. Confounding was controlled by including relevant covariates in the multivariable model. Results are presented as medians with interquartile ranges (IQRs), frequencies (n), percentages (%), PRs, and aPR with 95% confidence intervals (CI). A *p*-value <0.05 was considered statistically significant.

## 3. Results

### 3.1. Characteristics of Children

Table 1 shows the demographic and anthropometric characteristics of the 415 under-five children who were included in this study. Participants were selected through multistage sampling from PHC facilities in the Msinga sub-district of KwaZulu-Natal Province. The sex distribution showed a slightly higher proportion of girls (*n* = 230; 55%) compared to boys (*n* = 185; 45%). Regarding age distribution, 179 children (43%) were younger than 24 months, 133 (32%) were between 24 and 36 months, and 103 (25%) were older than 36 months. The median age was 24 months (interquartile range [IQR]: 12–24 months). Anthropometric assessments indicated a median weight of 8.7 kg (IQR: 6.9–10.2 kg), with values ranging from 3.6 kg to 16 kg. The median height/length was 75 cm (IQR: 65–83 cm), with a range of 60 cm to 111 cm. Nutritional status indicators showed a median WHZ/WLZ of −0.83 (range: −6.91 to 7.99) and a median MUACZ of −1.23 (range: −3.87 to 1.78), reflecting a wide spectrum of nutritional conditions among the children assessed.

### 3.2. Comparison of the Nutritional Indicators of Children

#### 3.2.1. Comparison by Sex

In Table 2, the nutritional status of under-five children, assessed using weight-for-height/length z-scores (WHZ/WLZ) and mid–upper-arm circumference-for-age z-scores (MUACZ) are compared by sex categories. Based on WHZ/WLZ, the prevalence of acute malnutrition was 29%, while 8% of children were classified as overweight or obese. Acute malnutrition was slightly more common among boys (30%) than girls (29%), and overweight/obesity was notably higher among boys (13%) compared to girls (5%). This difference in the distribution of WHZ/WLZ categories by sex was statistically significant (*p* = 0.014). Further categorization showed that 16% of children were moderately acutely malnourished (MAM) and 13% were severely acutely malnourished (SAM), with no significant sex differences observed (*p* = 0.418 and *p* = 0.668, respectively). According to MUACZ, 27% of children were acutely malnourished, with a slightly higher prevalence among boys (29%) than among girls (25%), though not statistically significant (*p* = 0.411). Among these, 18% were classified as MAM and 9% as SAM. While SAM was more prevalent among boys (12%) than among girls (6%), the difference was not statistically significant (*p* = 0.085).

#### 3.2.2. Comparison of Age Groups

Table 3 shows the nutritional status of under-five children, stratified by age group: ≤24 months, 24–36 months, and >36 months. Based on WHZ/WLZ, the prevalence of acute malnutrition increased significantly with age, from 20% in children ≤24 months to 28% in those aged 24–36 months, and 47% in children older than 36 months (*p* = 0.001). A similar trend was observed for SAM, which was most prevalent in the >36 months group (26%), compared to 9% in both the ≤24 months and 24–36 months groups (*p* =0.001). MAM also differed significantly by age, affecting 11% of children ≤24 months, 19% of those aged 24–36 months, and 21% of those >36 months (*p* = 0.037). In contrast, overweight/obesity was most common among children ≤ 24 months (16%), declined to 3% in the 24–36 months group, and was nearly absent in the >36 months group (1%) (*p* = 0.001). According to MUACZ, acute malnutrition was significantly more prevalent among older children, affecting 13% of those ≤ 24 months, 15% of those aged 24–36 months, and 58% of those > 36 months (*p* = 0.001). MAM and SAM also showed significant age-related differences: MAM was observed in 10%, 11%, and 36% of children in the ≤24 month, 24–36 month, and >36-month groups, respectively (*p* = 0.001), while SAM affected 3%, 4%, and 22% of children in the same age groups (*p* = 0.001).

### 3.3. Characteristics of Mothers

The demographic and socioeconomic characteristics of the 415 mothers of under-five children attending PHC facilities in the Msinga sub-district, KwaZulu-Natal Province, are summarized in Table 4. The median age of the mothers was 26 years (IQR: 22–31), with 47% aged below 25 years (Group 1; G1), 43% between 25 and 34 years (Group 2; G2), and 10% above 35 years (Group 3; G3). Many mothers were single, with significant differences in marital status observed across all age groups (*p* = 0.001), particularly between G1 and G2, G1 and G3, and G2 and G3 (all *p* = 0.001), completed at least secondary education, with a significant difference between G1 and G3 (*p* = 0.006) and between G2 and G3 (*p* = 0.003), and reported being unemployed (98%), with significantly more in G1 compared to G2 (*p* = 0.013). Household characteristics showed that 64% of mothers lived in parent-headed households, with significant differences across all age groups (*p* = 0.001). Most households (94%) had more than five members, with G1 significantly more likely to live in larger households compared to G2 (*p* = 0.001). Although the majority lived in brick houses, 18% resided in informal dwellings, with significant differences between G1 and G3 (*p* = 0.002). Access to electricity was high (96%), but significantly lower in G3 compared to G2 (*p* = 0.009). Use of firewood or coal as a primary energy source was common (46%), with significant differences between G1 and G2 (*p* = 0.012).

Table 5 presents selected variables on maternal obstetric history and selected infant feeding practices. Most mothers reported having one or two children, with parity significantly differing across all age groups (*p* = 0.001), particularly between G1 and G2, G1 and G3, and G2 and G3. Nearly all mothers (99%) delivered the participating child at full term in a hospital setting, although a significant difference was observed between G1 and G3 (*p* = 0.032). Obstetric complications were reported by 8% of mothers, with significantly higher rates in older age groups. The differences were significant across all comparisons (*p* = 0.001 for G1 vs G2 and G1 vs G3; *p* = 0.006 for G2 vs G3). BMI classifications also varied significantly by age group (*p* = 0.001). The mothers in G1 were more likely to have a normal BMI (18.5–24.99 kg/m^2^), while those in G2 and G3 had higher proportions of overweight individuals (25–29.9 kg/m^2^) and obesity (≥30 kg/m^2^), with a significant difference between G1 and G2 (*p* = 0.001). Breastfeeding was widely practiced (96%), though no significant differences were observed between groups. However, the duration of breastfeeding differed significantly between G1 and G2 (*p* = 0.001), with G2 more likely to continue breastfeeding beyond six months. Mixed feeding practices were reported by 27% of mothers, with significantly higher rates in G1 compared to both G2 (*p* = 0.01) and G3 (*p* = 0.042). Similarly, the early introduction of solid foods (before six months) was significantly more common in G1 than in G2 (*p* = 0.001).

### 3.4. Dietary Diversity and Food Group Consumption Among Children

In Table 6, a comparative analysis of the mean DDS and the proportions of children with normal and low DDS is presented, categorized by sex and age groups. The DDS among children ranged from 2.5 to 5 out of a possible eight food groups, with an overall mean of 3.65 ± 0.7. Statistically significant differences in mean DDS were observed between boys and girls (*p* = 0.030) and across different age groups (*p* < 0.001). While the prevalence of low DDS did not differ significantly by sex (*p* = 0.099), it varied notably across age categories (*p* < 0.001).

In the 24 h preceding the survey, the most consumed food groups among children were grains, roots, and tubers, consumed by 100% of the children. This was followed by vitamin A-rich fruits and vegetables (80%), dairy products (77%), and legumes (70%). In contrast, the consumption of flesh foods (21%), eggs (10%), other fruits and vegetables (10%), and other local foods (46%) was notably lower, with all falling below the 50% threshold (Figure 2).

### 3.5. Multivariate Analysis of Risk Factors for Acute Malnutrition

Table 7 shows the results of the multivariate regression analysis examining risk factors for acute malnutrition among children, incorporating variables with *p*-values < 0.25 from the univariate analysis to adjust for potential confounding. After adjusting for socio-demographic and health-related factors, several predictors remained significantly associated with acute malnutrition, as measured by WHZ/WLZ and MUACZ. For WHZ/WLZ, full-term pregnancy (aPR = 0.39; 95% CI: 0.23–0.64; *p* < 0.001), shorter breastfeeding duration (<3 months) (aPR = 0.59; 95% CI: 0.37–0.94; *p* = 0.028), and larger household size (≥5 members) (aPR = 0.66; 95% CI: 0.45–0.74; *p* = 0.035) were associated with lower risk. For MUACZ, mixed feeding was significantly associated with increased risk (aPR = 0.86; 95% CI: 0.55–1.18; *p* < 0.001), while higher dietary diversity (DDS ≥ 4) showed a protective association (aPR = 0.41; 95% CI: 0.04–0.84; *p* = 0.028). Children aged 36 months and older had significantly higher odds of malnutrition (aPR = 1.62; 95% CI: 1.15–2.10; *p* < 0.001). Maternal age was also associated with reduced risk, with children of mothers aged 25–34 years (aPR = 0.67; 95% CI: 0.48–0.93; *p* = 0.017) and ≥35 years (aPR = 0.58; 95% CI: 0.35–0.84; *p* = 0.043) experiencing significantly lower prevalence compared to those of mothers aged <25 years. Household income was strongly associated with acute malnutrition. Children from the lowest (<USD 280) and second lowest (USD 280–USD 560) income households had adjusted prevalence ratios approaching zero (aPR = 0.00 for both; *p* < 0.001), indicating a markedly higher risk compared to those from the highest income group (>USD 840). The USD 560–USD 840 group showed a non-significant trend toward reduced risk (aPR = 0.44; 95% CI: 0.07–2.75; *p* = 0.377).

## 4. Discussion

This study investigated the key risk factors associated with acute malnutrition in under-five children attending PHC facilities in the Msinga sub-district of KwaZulu-Natal Province, South Africa. It further assessed the socio-demographic, maternal and child factors that influence child nutritional outcomes. The findings contribute to the growing body of evidence on the dual burden of malnutrition in rural South Africa, highlighting the complex interconnections between inadequate dietary quality and child health.

The socio-demographic profile of participating mothers showed persistent structural inequities influencing child nutrition in South Africa. Most mothers were young, unmarried, and unemployed, living in overcrowded, low-income households and reliant on child support grants. These conditions, consistent with national trends, are linked to food insecurity and poor child nutritional outcomes [39,40,41], especially considering that social grants have been reported to be an unsustainable means of addressing poverty in the long term [20]. Furthermore, in SSA countries such as Ethiopia, children of single or illiterate mothers were found to be three times more likely to suffer from acute malnutrition, a pattern mirrored in South African contexts, where maternal agency is further undermined by systemic barriers and inadequate health system responsiveness [42,43]. Household dynamics, including the presence of extended family members such as grandparents, may influence maternal decision-making around child feeding, potentially shaping dietary diversity and caregiving practices [44,45]. These dynamics operate within a health system challenged by inconsistent implementation of SAM guidelines and limited clinical capacity, which may delay diagnosis and follow-up care [43]. This may affect early detection and treatment of acute malnutrition, particularly in resource-constrained settings. Furthermore, despite generally favourable obstetric profiles, with most births occurring at term in institutional settings, 8% of mothers reported complications such as preeclampsia and preterm delivery, conditions known to impair postnatal growth and IYCF practices, which, along with a reported 14.7% preterm birth rate, highlight the ongoing relevance of obstetric risks for child nutrition outcomes [46,47,48].

Our findings, in the context of ongoing national nutrition interventions during the study period, show that despite the presence of programs such as the INP and CSG [15,20], significant gaps remain in addressing acute malnutrition in rural areas such as the study area. These programs, while designed to improve child nutrition through growth monitoring, education, and financial support, often face implementation challenges, such as delayed food deliveries and limited community outreach. Additionally, the CMAM strategy, including the use of RUTF [15], was inconsistently applied across facilities, potentially limiting its impact. These contextual factors may have shaped both dietary patterns and the prevalence of acute malnutrition observed in this study.

In this study, 27% of mothers practiced mixed feeding, and an equal proportion introduced complementary foods before six months, both of which are suboptimal IYCF practices linked to increased risk of infection and malnutrition [49,50,51]. Similar trends have been reported in Ethiopia, where early complementary feeding and prelacteal feeding remain common [42], often driven by cultural beliefs, misconceptions about breast milk sufficiency, and limited maternal nutrition education. Dietary diversity was also limited, with near-universal reliance on starchy staples and low intake of nutrient-dense foods such as flesh foods (21%), eggs (10%), and non-vitamin A-rich fruits and vegetables (10%), reflecting persistent food insecurity [52,53,54]. These patterns are consistent with regional data showing that only 11% of children in sub-Saharan Africa meet MDD, with even lower rates in East Africa [55,56]. Dietary diversity was assessed using the WHO-recommended 24 h recall method, which is practical and widely used in similar settings. While it offers a standardized snapshot of recent intake, we acknowledge its limitations in capturing habitual dietary patterns. The 24 h recall was selected for its feasibility in field settings and alignment with global IYCF indicators. Our choice was based on feasibility and alignment with global IYCF indicators.

The prevalence of acute malnutrition in this study remains high, with rates of 29% based on WHZ/WLZ and 27% based on MUACZ. A similar prevalence has been reported locally [43] and in other parts of Africa [42]. However, higher rates have been observed in regions such as Ethiopia (42%) [57], Nigeria (35%) [58], and some Asian countries such as Pakistan (46%) [59]. Within South Africa, prevalence varies significantly across provinces, ranging from 1% to 22% [60], with a national average of 2.5% [61]. Furthermore, in this study, the prevalence of MAM ranged between 16% and 17%, while SAM ranged from 9% to 13%. In other parts of the country, such as a rural setting in Western Cape, 36% of children admitted to a health facility presented with SAM [48]. Although South Africa’s SAM case fatality rate (CFR) has declined modestly from 7.7% to 7%, disparities persist across provinces. Notably, uMzinyathi District, where Msinga is located, reported a CFR of 9%, exceeding the national average [17]. At the African continent level, a systematic review of 52 studies reported an average under-five mortality rate of 11% due to SAM [62]. These findings highlight the persistent burden of undernutrition, driven by chronic food insecurity, inadequate dietary intake, and suboptimal caregiving. The higher rates of wasting among children aged 36 months and older further suggest cumulative nutritional deficits over time [63,64,65].

The coexistence of undernutrition and overweight, particularly among boys, reflects South Africa’s ongoing nutrition transition [66,67,68]. In our study, overweight and obesity were significantly more prevalent among boys (13%) than among girls (5%), a pattern consistent with findings from other South African studies, such as Pienaar (2015) [69], who reported higher overweight rates among boys in rural Free State, and Sharma et al. (2024) [70], who observed similar disparities among preschoolers in Gauteng. Regionally, a large-scale analysis of 33 sub-Saharan African countries found that under-five male children had significantly higher odds of being overweight or obese compared to females [71], with South Africa showing one of the highest prevalence rates in the region. For example, the studies from Ethiopia, Nigeria, and Kenya included in this review reported higher overweight prevalence among boys, with Ethiopia showing a gender disparity in urban settings, Nigeria highlighting cultural feeding practices favouring boys, and Kenya noting higher BAZ among male preschoolers [71]. This disparity may be attributed to biological differences in growth velocity and fat distribution [72], as well as sociocultural feeding practices that favour boys. Moreover, the increasing availability of energy-dense, nutrient-poor foods, even in food-insecure households, contributes to this dual burden of malnutrition [30,39,73].

Multivariate analysis identified several significant predictors of acute malnutrition using both WHZ/WLZ and MUACZ. Protective factors included full-term pregnancy, shorter breastfeeding duration (<3 months), larger household size (≥5 members), and adequate dietary diversity. Mixed feeding was significantly associated with lower risk, although the confidence interval approached 1.00, suggesting a modest effect size. This finding contrasts with previous literature that often associates mixed feeding with increased risk and may reflect contextual factors such as food availability or maternal support. Full-term pregnancy showed a strong protective effect, confirming the importance of maternal and perinatal health [74,75]. Children from households with five or more members were less likely to be acutely malnourished, possibly due to more focused caregiving or reduced competition for resources. Breastfeeding for less than three months was associated with increased risk, emphasizing the benefits of sustained breastfeeding [74,75]. Maternal ages of 25–34 years and ≥35 years were also protective, potentially reflecting greater caregiving capacity and household stability [76]. Is has been suggested that mothers aged 25 years and older are more likely to provide stable and nurturing environments for their children, benefiting from accumulated life experience, better decision-making skills, and often improved socioeconomic conditions, all of which contribute to reduced risk of acute malnutrition and better overall child health outcomes [77]. Children with a dietary diversity score of four or more had better MUACZ scores, indicating improved nutrition, consistent with findings from other African settings [11,62,78,79]. This is because a varied diet provides essential nutrients that support growth and development, and is recognized by WHO as a key indicator of diet quality in young children [79,80].

Conversely, risk factors included older child age (≥36 months), low household income, and mixed feeding practices. Children aged 36 months and older were more vulnerable, likely due to cumulative nutritional deficits, as reported in regional studies from sub-Saharan Africa [11,62,78]. Low household income was strongly associated with malnutrition, aligning with evidence from South Africa, Ethiopia, and Bangladesh [60,76,81,82], and is consistent with the UNICEF (2023) conceptual framework [26], which identifies poverty, food insecurity, and inadequate care as key drivers of malnutrition. Children from the lowest (<USD 280) and second lowest (USD 280–USD 560) income households had adjusted prevalence ratios approaching zero, indicating a markedly higher risk compared to those from the highest income group. The USD 560–USD 840 income group showed a non-significant trend toward reduced risk. Mixed feeding and early introduction of complementary foods were also associated with increased risk, likely due to reduced breast milk intake and heightened exposure to infections [83,84]. These findings highlight the multifactorial nature of acute malnutrition and the need for integrated interventions that address maternal health, infant feeding practices, and socioeconomic conditions.

This study has several limitations that should be considered. First, the cross-sectional design limits causal inference and generalizability beyond the rural setting, despite the use of multistage sampling. The absence of maternal dietary data restricts exploration of intergenerational nutrition. This study focused on key IYCF indicators relevant to the context, did not apply the full WHO framework, and lacked nutrient intake data and biomarkers, limiting the depth of assessment. Second, oedema was not assessed, as this study focused on dietary diversity and anthropometric indicators rather than clinical diagnostics. While WHZ/WLZ, MUACZ, and WAZ are sufficient for identifying malnutrition in population-based studies, the absence of oedema may limit detection of some SAM cases. Third, although the sample size was sufficient, exclusion of five cases due to missing data may introduce minor bias, though the 100% participation rate reduces the risk of selection bias. Lastly, while the 24 h recall is widely used and suited to low-resource settings, it may not reflect the usual intake due to daily variation. The 7-day recall, particularly when used prospectively, can offer a broader dietary picture, though it also presents challenges such as increased respondent burden. Our selection was based on contextual feasibility and alignment with global IYCF standards. Despite these limitations, this study offers valuable insights into child nutrition in rural South Africa. Future research should adopt longitudinal designs and include maternal intake and clinical biomarkers.

## 5. Conclusions

This study showed a dual burden of undernutrition and overweight, indicating the complexity of South Africa’s rural nutrition landscape. Acute malnutrition was strongly linked to poor dietary quality, marked by limited dietary diversity, early complementary feeding, and dependence on starchy staples, despite high breastfeeding rates. Chronic food insecurity further constrained access to nutrient-dense foods. The key risk factors identified in the multivariate analysis included child age, low household income, mixed feeding practices, shorter breastfeeding duration, and younger maternal age (<25 years). Protective factors included full-term pregnancy, maternal age 25–34 and ≥35 years, larger household size (≥5 members), and adequate dietary diversity. These findings contribute to the growing body of evidence on the complex drivers of acute malnutrition in rural South Africa. They offer valuable insights for tailoring community-based nutrition strategies. Importantly, the results should be interpreted considering ongoing national interventions such as the Integrated Nutrition Programme, the Child Support Grant, and the Community-based Management of Acute Malnutrition, which were active during the study period. While these programs aim to improve child nutrition, their uneven implementation may have influenced both dietary practices and malnutrition outcomes observed in this setting. Moreover, although social grants provide essential short-term relief, they have been reported not to offer a sustainable solution to poverty, which remains a key structural driver of malnutrition. Strengthening the reach and effectiveness of these interventions remains critical to achieving sustainable improvements in child health and nutrition.

## Figures and Tables

**Figure 1 nutrients-17-02038-f001:**
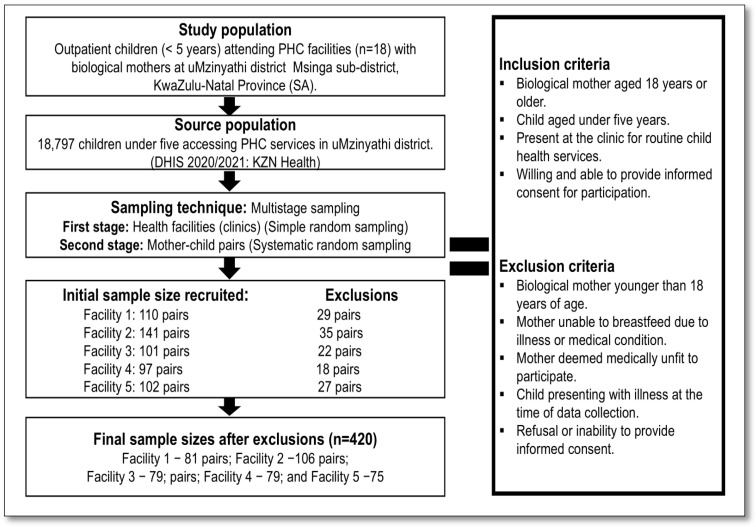
The recruitment process for this study.

**Figure 2 nutrients-17-02038-f002:**
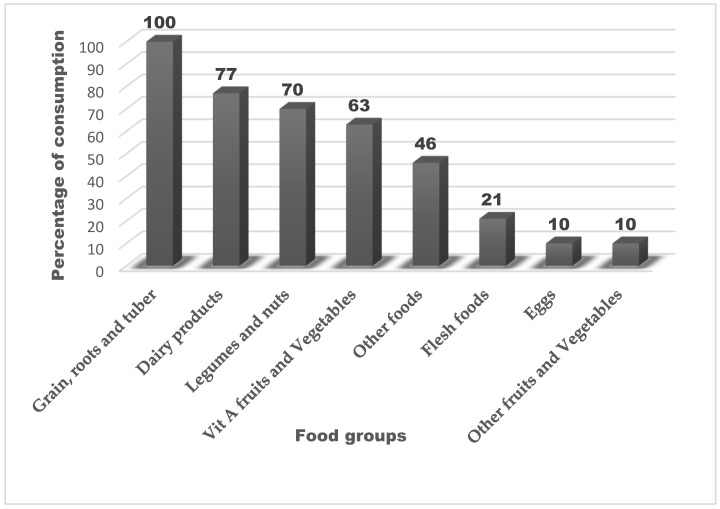
Food group consumption patterns among children.

**Table 1 nutrients-17-02038-t001:** Age, anthropometric, and nutritional indicators of children.

Variables	Median	IQR	Minimum	Maximum
Age (months)	24	12;24	12	60
Weight (kg)	8.3	6.9; 10.2	3.6	16
Length/height (cm)	73	65; 83	48	116
WHZ/WLZ	−0.89	−2.2; 0.41	−6.91	7.99
MUACZ	−1.16	−2.1; −0.4	−3.87	1.78

Descriptive statistics for continuous variables include median, interquartile range (IQR: 25th and 75th percentiles), minimum, median, and maximum. WHZ/WLZ stands for weight-for-height or weight-for-length z-scores, and MUACZ for mid–upper-arm circumference z-scores.

**Table 2 nutrients-17-02038-t002:** Comparison of nutritional indicators of children by sex.

Variables	All	Boys	Girls	
	**n (%)**	**n (%)**	**n (%)**	** *p* ** **-Value**
WHZ/WLZ				0.014 *
Normal (≥−2SD)	252 (62)	104 (57)	148 (63)	
Acute malnutrition (<−2 SD)	118 (29)	54 (30)	64 (29)	0.415
MAM (between −3 SD and −2 SD)	64 (16)	30 (17)	34 (15)	0.418
SAM (<−3SD)	54 (13)	24 (13)	30 (14)	0.668
Overweight/obesity (>+2SD)	34 (8)	23 (13)	11 (5)	0.004 *
MUACZ				0.411
Normal (≥−2SD)	251 (73)	101 (71)	15 (75)	
Acute malnutrition (<−2 SD)	93 (27)	42 (29)	51 (25)	0.411
MAM (between −3 SD and −2 SD)	63 (18)	25 (17)	38 (19)	0.936
SAM (<−3SD)	30 (9)	17 (12)	13 (6)	0.085

* Indicates a statistically significant difference; acute malnutrition was classified according to the WHO growth standards. For weight-for-height/length z-scores (WHZ/WLZ), acute malnutrition was defined as <−2 standard deviations (SDs), and overweight/obesity as >+2 and +3 SD. For mid–upper-arm circumference-for-age z-scores (MUACZ), acute malnutrition was defined as <−2 SD. The Chi-squared test (χ^2^) was used.

**Table 3 nutrients-17-02038-t003:** Comparison of nutritional indicators of children by age.

Variables	≤24 Months	24–36 Months	>36 Months	
	n (%)	n (%)	n (%)	*p*-Value
WHZ/WLZ				0.001 *
Normal (≥−2SD)	114 (64)	87 (69)	51 (51)	
Acute malnutrition (<−2 SD)	36 (20)	35 (28)	47 (47)	0.001 *
MAM (between −3 SD and −2 SD)	19 (11)	24 (19)	21 (21)	0.037 *
SAM (<−3SD)	17 (9)	11 (9)	26 (26)	0.001 *
Overweight/obesity (>+2SD)	29 (16)	4 (3)	1 (1)	0.001 *
MUACZ				0.001 *
Normal (≥−2SD)	99 (89)	110 (85)	42 (42)	
Acute malnutrition (<−2 SD)	15 (13)	20 (15)	58 (58)	0.001 *
MAM (between −3 SD and −2 SD)	12 (10)	15 (11)	36 (36)	0.001 *
SAM (<−3SD)	3 (3)	5 (4)	22 (22)	0.001 *

* Indicates a statistically significant difference; acute malnutrition was classified according to the WHO growth standards. For weight-for-height/length z-scores (WHZ/WLZ), acute malnutrition was defined as <−2 standard deviations (SDs), and overweight/obesity as >+2 and +3 SD. For mid–upper-arm circumference-for-age z-scores (MUACZ), acute malnutrition was defined as <−2 SD. The Chi-squared test (χ^2^) was used.

**Table 4 nutrients-17-02038-t004:** Comparison of mothers’ sociodemographic characteristics.

Variables	All n (%)	<25 Years n (%)	25–34 Years n (%)	≥35 Years n (%)	*p*-Value
Marital status					
Single	347 (84)	187 (97)	143 (79)	17 (40)	P^0^ = 0.001 *; P^1^ = 0.001 * P^2^ = 0.001 *; P^3^ = 0.001 *
Ever married	68 (16)	6 (3)	37 (21)	25 (60)	
Level of education					
No school/Primary	11 (3)	3 (2)	3 (2)	5 (12)	P^0^ = 0.005 *
Secondary	151 (36)	73 (38)	63 (35)	15 (36)	P^1^ = 0.130
Completed grade 12.	231 (56)	111 (57)	98 (54)	22 (52)	P^2^ = 0.006 *
Post grade 12	22 (5)	6 (3)	16 (9)	0	P^3^ = 0.003 *
Employed					
No	410 (98)	191 (99)	170 (94)	40 (95)	P^0^ = 0.026 *; P^1^ = 0.013 * P^2^ = 0.091; P^3^ = 0.838
Yes	5 (2)	2 (9)	10 (6)	2 (5)	
Receiving child grant					
No	64 (15)	29 (15)	170 (94)	50 (95)	P^0^ = 0.223; P^1^ = 0.473 P^2^ = 0.177; P^3^ = 0.089
Yes	351 (85)	164 (85)	10 (6)	2 (5)	
Household head					
Spouse/partner.	64 (15)	6 (3)	33 (18)	25 (60)	P^0^ = 0.001 *
Parents	267 (64)	146 (76)	105 (58)	16 (38)	P^1^ = 0.001 *
Grandparents	79 (19)	39 (20)	39 (22)	1 (2)	P^2^ = 0.001 *
Other relatives	5 (2)	2 (1)	3 (2)	0	P^3^ = 0.001 *
Household size					
≤4	25 (6)	4 (2)	19 (11)	2 (5)	P^0^ = 0.002 *; P^1^ = 0.001 * P^2^ = 0.317; P^3^ = 0.248
≥5	390 (94)	189 (98)	161 (89)	40 (95)	
Household income					
<USD 280	391 (94)	185 (96)	165 (92)	41 (98)	P^0^ = 0.471
USD 280–USD 560	11 (3)	1 (2)	5 (3)	1 (2)	P^1^ = 0.001 *
USD 560–USD 840	6 (1)	1 (1)	5 (3)	0	P^2^ = 0.317
>USD 840	7 (2)	2 (1)	5 (2)	0	P^3^ = 0.248
Dwelling place					
RDP house	4 (1)	3 (2)	1 (1)	0	P^0^ = 0.002 *; P^1^ = 0.819; P^2^ = 0.053; P^3^ = 0.002 *
Shack/mud house	74 (18)	49 (25)	21 (11)	4 (10)	
Brick house	337 (81)	141 (73)	158 (88)	38 (90)	
Access to electricity					
No	17 (4)	10 (5)	3 (2)	4 (10)	P^0^ = 0.029 *; P^1^ = 0.064; P^2^ = 0.281; P^3^ = 0.009 *
Yes	398 (96)	183 (95)	177 (98)	38 (90)	
Refrigerator use					
No	64 (15)	34 (18)	22 (12)	8 (19)	P^0^ = 0.280; P^1^ = 0.145; P^2^ = 0.826; P^3^ = 0.244
Yes	351 (85)	159 (82)	158 (88)	34 (81)	
Source of energy					
Firewood/coal	191 (46)	100 (52)	69 (38)	22 (53)	P^0^ = 0.013 *; P^1^ = 0.012 *; P^2^ = 0.896; P^3^ = 0.168
Paraffin/gas	6 (1)	1 (1)	5 (3)	0	
Electricity	218 (53)	92 (47)	106 (59)	20(47)	
Access to water					
No	398 (96)	187 (97)	169 (94)	42 (100)	P^0^ = 0.159; P^1^ = 0.165; P^2^ = 0.247; P^3^ = 0.100
Yes	17 (4)	6 (3)	11 (6)	0	

USD denotes US dollars, converted from South African rands (ZAR) at an exchange rate of 1 ZAR = 0.056 USD. P stands for probability level; G1, G2, and G3 stand for Group 1, Group 2, and Group 3, respectively. P^0^: *p*-value for comparison among the 3 Groups; P^1^: comparison between G1 and G2; P^2^: between G1 and G3; P^3^: between G2 and G3; * indicates significant differences. The Chi-squared test (χ^2^) was used.

**Table 5 nutrients-17-02038-t005:** Comparison of mothers’ obstetric history and selected infant feeding practices.

Variables	All n (%)	<25 (G1) n (%)	25–34 (G2) n (%)	≥35 (G3) n (%)	*p*-Value
BMI (kg/m^2^)					
<18.5 (underweight)	22 (5)	6 (3)	13 (7)	3 (7)	P^0^ = 0.001 *
>18.5–24.99 (normal)	286 (69)	158 (82)	99 (55)	29 (69)	P^1^ = 0.001 *
25–29.9 (overweight)	74 (18)	27 (14)	41 (23)	6 (14)	P^2^ = 0.008
≥30 (obesity)	33 (8)	2 (1)	27 (15)	4 (10)	P^3^ = 0.283
Parity					
1–2	350 (84)	191 (99)	148 (82)	11 (26)	P^0^ = 0.001 *; P^1^ = 0.001 *; P^2^ = 0.001 *; P^3^ = 0.001 *
≥3	65 (16)	2 (1)	32 (18)	31 (74)	
Pregnancy full term					
No	3 (1)	0	2 (1)	1 (2)	P^0^ = 0.085; P^1^ = 0.142; P^2^ = 0.032 *; P^3^ = 0.521
Yes	412 (99)	193 (100)	178 (99)	41 (98)	
Obstetric complications					
No	379 (91)	190 (98)	157 (88)	30 (71)	P^0^ = 0.001 *; P^1^ = 0.001 *; P^2^ = 0.001 *; P^3^ = 0.006
Yes	36 (8)	3 (2)	21 (12)	12 (29)	
Child breastfed					
No	16 (3)	6 (3)	9 (5)	1 (2)	P^0^ = 0.650; P^1^ = 0.353; P^2^: n/a; P^3^ = 0.628
Yes	399 (96)	187 (97)	17 (5)	41 (98)	
Duration of breastfeeding					
<3 months	43 (10)	32 (17)	9 (5)	2 (5)	P^0^ = 0.001 *
3–6 months	69 (17)	36 (18)	27 (15)	6 (14)	P^1^ = 0.001 *
>6 months	242 (48)	91 (47)	123 (68)	28 (67)	P^2^ = 0.089
Continued	61 (15)	34 (18)	21 (12)	6 (14)	P^3^ = 0.974
Mixed feeding					
No	303 (73)	125 (65)	144 (80)	34 (81)	P^0^ = 0.002 *; P^1^ = 0.001 *; P^2^ = 0.042 *; P^3^ = 0.889
Yes	112 (27)	68 (35)	36 (20)	8 (19)	
Introduction of solid foods					
Not yet	65 (16)	34 (18)	22 (12)	9 (21)	P^0^ = 0.001 *; P^1^ = 0.001 *; P^2^ = 0.126; P^3^ = 0.285
<6 months	110 (26)	68 (35)	34 (19)	8 (19)	
>6 months	240 (58)	91 (47)	124 (69)	25 (60)	

P stands for probability level; G1, G2, and G3 stand for Group 1, Group 2, and Group 3, respectively. P^0^: *p*-value for comparisons among the 3 groups; P^1^: between G1 and G2; P^2^: between G1 and G3; P^3^: between G2 and G3; * indicates a statistically significant difference; n/a shows that the test was not computed. The Chi-squared test (χ^2^) was used.

**Table 6 nutrients-17-02038-t006:** Comparison of means and proportions of DDS among children by sex and age.

**Variables**	**All**	**Boys**	**Girls**	** *p-* ** **Value**
DDS				
Mean	3.65 ± 0.7	3.56 ± 0.69	3.71 ± 0.63	
Normal	340 (82)	158 (85)	182 (79)	0.030 *
Low	75 (18)	27 (15)	48 (21)	0.099
**Variables**	**<24 Months**	**24–36 Months**	**≥36 Months**	** *p-* ** **Value**
DDS				
Mean	3.39 ± 0.71	3.79 ± 0.52	3.89 ± 0.58	
Normal	159 (89)	108 (81)	73 (71)	<0.001 *
Low	20 (11)	25 (19)	30 (29)	<0.001 *

DDS stands for dietary diversity scores; the Kruskal–Wallis (Wilcoxon rank-sum) test was used; * indicates a statistically significant difference.

**Table 7 nutrients-17-02038-t007:** Risk factors for acute malnutrition among children.

Variables	PR (95%CI)	*p*-Value	aPR (95%CI)	*p*-Value
Acute malnutrition (WHZ/WLZ)				
Marital status				
Single	1 (ref)		1 (ref)	
Married	1.28 (0.53–1.15)	0.202	(0.50–1.08)	0.114
Number of household members				
<5	1 (ref)		1 (ref)	
≥5	0.65 (0.45–0.94)	0.024 *	0.66 (0.45–0.74)	0.035 *
Full-term pregnancy				
No	1 (ref)		1 (ref)	
Yes	0.37 (0.33–0.42)	<0.001 *	0.39 (0.23–0.64)	<0.001 *
Duration of breastfeeding				
Still breastfeeding	1 (ref)		1 (ref)	
<3 months	0.61 (0.38–1.03)	0.039 *	0.59 (0.37–0.94)	0.028 *
months	0.74 (0.52–1.05)	0.088	0.77 (0.54–1.09)	0.148
>6 months	0.79 (0.51–1.22)	0.280	0.79 (0.50–1.25)	0.302
Acute malnutrition (MUACZ)				
DDS				
<4	1 (ref)		1 (ref)	
≥4	0.46 (0.09–0.83)	0.016 *	0.41 (0.04–0.8378	0.028 *
Child’s age (months)				
<24	1 (ref)		1 (ref)	
24–36	1.17 (0.63–2.18)	0.622	1.34 (0.75–2.36)	0.329
≥36	1.48 (0.98–1.98)	<0.001 *	1.62 (1.15–2.10)	<0.001 *
Mother’s age (years)				
<25	1 (ref)		1 (ref)	
25–34	0.63 (0.43–0.90)	0.013 *	0.67 (0.48–0.93)	0.017 *
≥35	0.67 (0.35–1.28)	0.230	0.58 (0.35–0.84)	0.043 *
Household income				
<USD 280	1 (ref)		1 (ref)	
USD 280–USD 560	0.00 (0.00–0.00)	<0.001 *	0.00 (0.00–0.00)	<0.001 *
USD 560–USD 840	0.00 (0.00–0.00)	<0.001 *	0.00 (0.00–0.00)	<0.001 *
>USD 840	0.71 (0.12–4.14)	0.702	0.44 (0.07–2.75)	0.377
Mixed feeding				
No	1 (ref)		1 (ref)	
Yes	0.84 (0.51–1.18)	<0.001 *	0.86 (0.55–1.18)	<0.001 *

* Indicates a statistically significant association *(p* <0.05). Poisson regression with robust standard errors was used. USD denotes US dollars, converted from South African rands (ZAR) at an exchange rate of 1 ZAR = 0.056 USD.

## Data Availability

Due to ethical considerations, the dataset generated and analysed for the study group during this research is not publicly available but can be obtained from the corresponding author upon a reasonable request.

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
