# Peer review of "Acute Malnutrition in Under-Five Children in KwaZulu-Natal, South Africa: Risk Factors and Implications for Dietary Quality"

_nutrients, 2025, doi:10.3390/nu17122038_

Round 1

Reviewer 1 Report

Comments and Suggestions for Authors

In abstract

in sentence The prevalence of acute malnutrition was 20% based on WHZ/WLZ and 27% based on MUACZ, with 17% of children classified as severely malnourished. - it is unclear if there were 17 % severe malnourished from the whole sample or within the malnourished group, please clarify

please rephrase to under-five children, also in keywords

there is a space missing in line 93 and 207

line 149 missing period

line 152 extra space, Table 5 parity and solids introduction, line 372

In tables please include statistical test used in the footnotes

In line 289, all p values should be rounded at 3 decimal spaces, also in tables

Can you present income also in USD or just give reference to the readers?

lines 317 to 330  please change formatting

Can you please discuss on the possible reasons for male children being more often overweight? there is very little on your own results in the discussion section

Author Response

REVIEWER 1

In abstract in sentence The prevalence of acute malnutrition was 20% based on WHZ/WLZ and 27% based on MUACZ, with 17% of children classified as severely malnourished. - it is unclear if there were 17 % severe malnourished from the whole sample or within the malnourished group, please clarify.

Response: Thank you for this insightful comment. Based on your suggestion and following a re-analysis of the data in consultation with our biostatistician, we have revised the abstract to report only the overall prevalence of acute malnutrition: 29% based on WHZ/WLZ and 27% based on MUACZ. The 17% figure previously mentioned referred to the proportion of children classified as severely malnourished within the malnourished group, not the entire sample. This has now been clarified in the Results section (Table 2, lines 335–336) - after re-analysis. 

please rephrase to under-five children, also in keywords

Response: The term has been revised to “under-five children” throughout the manuscript, including in the keywords section

there is a space missing in line 93 and 207

Response: The missing spaces in previous lines 93 (shifted to line 175) and 207 (shifted to line 242) have been corrected, and the manuscript has been carefully reviewed to ensure consistency in formatting throughout.

line 149 missing period

Response: we have added period “Data were collected from November 2022 to May 2023” shifted to line 178.

line 152 extra space, Table 5 parity and solids introduction, line 372

Response: Extra spaces have been corrected, line 152 shifted to line 181), and  in table 5; lines 408 – 409.

In tables please include statistical test used in the footnotes

Response: All the statistical tests in Tables 1 (line 311); 2 (line 338); 3 (line 361); 4 (line 384); 5 (lines 412); 6 (line 422) and 7 (line 455)

In line 289, all p values should be rounded at 3 decimal spaces, also in tables

Response: All p-values have been rounded off to 3 decimal spaces. Those that were ≤0.0001 are now <0.001.

Can you present income also in USD or just give reference to the readers?

Response: Income has been converted to USD in Tables 4 (lines 380 – 381) and 7 (lines 452 – 453). Explanation of USD to ZAR has been provided in the footnote of both tables

lines 317 to 330  please change formatting

Response: The formatting of lines 317 to 330 has been updated to align with the MDPI author guidelines: Palatino Linotype, font size 10, line spacing set to “at least,” with 12 pt before and 6 pt after.

Can you please discuss on the possible reasons for male children being more often overweight? there is very little on your own results in the discussion section

Response: Discussed in lines 511 - 527

Reviewer 2 Report

Comments and Suggestions for Authors

Report 1

Hello, thank you for the opportunity to evaluate this article.

I believe that the article is original, and the information  is presented in a clear manner, respecting scientific rigor.

I would like to make a few suggestions to the authors:

  1. In the discussion section, I suggest comparing the results obtained with those from other similar studies on the population of South Africa or other regions, highlighting the particularities observed in your study.

For example:

Mihret ST, Biset G, Nurye NA. Prevalence of Acute Malnutrition and Associated Factors among Children aged 6-59 months in South Wollo Zone, East Amhara, Northeast Ethiopia: a Community-based cross-sectional study. BMJ Open. 2023 Oct 24;13(10):e062582. doi: 10.1136/bmjopen-2022-062582. PMID: 37879690; PMCID: PMC10603487.

Mambulu-Chikankheni FN. Factors influencing the implementation of severe acute malnutrition guidelines within the healthcare referral systems of rural subdistricts in North West Province, South Africa. PLOS Glob Public Health. 2023 Aug 18;3(8):e0002277. doi: 10.1371/journal.pgph.0002277. PMID: 37594922; PMCID: PMC10437970.

Thomas A, Engelbrecht AL, Slogrove AL. Severe acute malnutrition outcomes for children of South African compared to foreign-born parents admitted to a rural regional hospital in South Africa: a retrospective cohort study. J Trop Pediatr. 2022 Oct 6;68(6):fmac097. doi: 10.1093/tropej/fmac097. PMID: 36350713; PMCID: PMC9645353.

Wambua, J., Ali, A., Ukwizabigira, J.B. et al. Prevalence and risk factors of under-five mortality due to severe acute malnutrition in Africa: a systematic review and meta-analysis. Syst Rev 14, 29 (2025). https://doi.org/10.1186/s13643-024-02740-9

2.You stated: The observed protective association between maternal obesity and child nutritional status may indicate improved household food access or greater maternal health literacy; however, this relationship warrants further investigation. (lines 424-426)

I suggest you include a paragraph briefly detailing how maternal obesity may influence children's nutritional status.

Author Response

REVIEWER 2

Hello, thank you for the opportunity to evaluate this article.

I believe that the article is original, and the information  is presented in a clear manner, respecting scientific rigor. I would like to make a few suggestions to the authors: 1. In the discussion section, I suggest comparing the results obtained with those from other similar studies on the population of South Africa or other regions, highlighting the particularities observed in your study.

Response: Thank you for suggesting these papers. We have read them and found them very relevant to use for our discussion. The entire discussion has change after we re-analysed data to get the adjusted prevalence ratio.

For example:Mihret ST, Biset G, Nurye NA. Prevalence of Acute Malnutrition and Associated Factors among Children aged 6-59 months in South Wollo Zone, East Amhara, Northeast Ethiopia: a Community-based cross-sectional study. BMJ Open. 2023 Oct 24;13(10):e062582. doi: 10.1136/bmjopen-2022-062582. PMID: 37879690; PMCID: PMC10603487.

Mambulu-Chikankheni FN. Factors influencing the implementation of severe acute malnutrition guidelines within the healthcare referral systems of rural subdistricts in North West Province, South Africa. PLOS Glob Public Health. 2023 Aug 18;3(8):e0002277. doi: 10.1371/journal.pgph.0002277. PMID: 37594922; PMCID: PMC10437970.

Thomas A, Engelbrecht AL, Slogrove AL. Severe acute malnutrition outcomes for children of South African compared to foreign-born parents admitted to a rural regional hospital in South Africa: a retrospective cohort study. J Trop Pediatr. 2022 Oct 6;68(6):fmac097. doi: 10.1093/tropej/fmac097. PMID: 36350713; PMCID: PMC9645353.

Wambua, J., Ali, A., Ukwizabigira, J.B. et al. Prevalence and risk factors of under-five mortality due to severe acute malnutrition in Africa: a systematic review and meta-analysis. Syst Rev 14, 29 (2025). https://doi.org/10.1186/s13643-024-02740-9

2.You stated: The observed protective association between maternal obesity and child nutritional status may indicate improved household food access or greater maternal health literacy; however, this relationship warrants further investigation. (lines 424-426). I suggest you include a paragraph briefly detailing how maternal obesity may influence children's nutritional status.

Response: Following reanalysis using Prevalence Ratios instead of Odds Ratios, we found no statistically significant association between maternal BMI and child acute malnutrition indicators (WHZ/WLZ and MUACZ) in the adjusted models. As a result, we did not expand on the potential mechanisms linking maternal obesity to child nutritional status in the revised manuscript, as the data did not support further interpretation.  

Reviewer 3 Report

Comments and Suggestions for Authors

The manuscript entitled “nutrients-3672362_ Acute Malnutrition in Under-Five Children in KwaZulu-Natal, South Africa: Risk Factors and Implications for Dietary Quality” is submitted for potential publication in the “Nutrition and Public Health” of the special issue “Food Insecurity, Nutritional Status, and Human Health”. The study addresses the persistent problem of acute malnutrition among children under five years of age in rural KwaZulu-Natal, South Africa, despite ongoing national efforts. Based on a cross-sectional analysis of 415 mother–child pairs, the study found that 20–27% of children were malnourished, with 17% being severely affected. Low dietary diversity was observed in 82% of the sample. Malnutrition was associated with child age, maternal marital status, and maternal obesity. These findings emphasize the urgent need for integrated, household-level interventions to improve nutritional outcomes and infant and young child feeding (IYCF) practices. Therefore, the content of the study is appropriate for inclusion in this volume.

In the methodology section of the abstract, the authors indicate that a cross-sectional study was conducted on 415 mother–child pairs. However, it is important to specify the source population from which this sample was drawn, as well as the participation rate.

The introduction appropriately outlines the significance of severe malnutrition among children under five in Africa, supported by relevant literature. However, given that the study concludes with a call for “integrated, household-level strategies to strengthen social protection and improve dietary diversity and IYCF practices,” the introduction should explicitly describe which such strategies were in place at the time of the study. This contextual information is necessary to assess the implications of the findings.

Furthermore, the final part of the introduction should clearly articulate the research hypothesis and the specific study objective. Currently, this section is confusing, and I recommend it be rewritten with particular attention to stating a clear and precise study objective.

In the Materials and Methods section, the study design is described as cross-sectional, conducted between 2022 and 2023. However, the manuscript does not describe the underlying population in the study area, which reportedly includes 18,797 children under the age of five. Ethical approval and informed consent procedures from parents or guardians should be explicitly documented.

The sample size calculation assumes a 10% non-participation rate; however, the actual participation rate should be reported. Similarly, in Figure 1, the origin of the source population should be clearly indicated.

In the data collection section, it is important to specify who collected the information and who performed anthropometric measurements on the children. The dietary questionnaire used was a 24-hour recall, which should be critically discussed, as this method is prone to bias and is generally considered less reliable than a seven-day recall.

While the procedures for measuring maternal and child anthropometry are described, it remains unclear who performed the measurements.

The statistical analysis section should detail the measures used to describe the results, particularly the prevalence. Since this is a cross-sectional design, prevalence ratios are more appropriate than odds ratios. Both crude and adjusted ratios should be presented to assess whether covariates modify the effect.

Tables 2 and 3 show the distribution of children by sex and nutritional status. These tables should include the criteria used to classify malnutrition, and this information should also be clearly stated in the Materials and Methods section. The p-value presented must indicate whether it refers to comparisons by sex or between the “yes” and “no” categories of each nutritional status group, to ensure the table is self-explanatory.

In Table 4, statistical tests should be used to compare the different variable categories presented.

In Table 7, both crude and adjusted risks should be reported, rather than only the adjusted ones.

The discussion contextualizes the findings with relevant literature, acknowledging the limitations inherent to a cross-sectional design, which only allows for the identification of potential risk factors. However, the authors should be more cautious when drawing conclusions.

The conclusion should not merely summarize the study but rather provide a direct answer to the study objective.

In summary, the manuscript presents valuable insights into the nutritional status of mothers and children under five in a rural South African setting. However, several aspects require clarification and revision before the manuscript can be considered for publication.

Author Response

REVIEWER 3

The manuscript entitled “nutrients-3672362_ Acute Malnutrition in Under-Five Children in KwaZulu-Natal, South Africa: Risk Factors and Implications for Dietary Quality” is submitted for potential publication in the “Nutrition and Public Health” of the special issue “Food Insecurity, Nutritional Status, and Human Health”. The study addresses the persistent problem of acute malnutrition among children under five years of age in rural KwaZulu-Natal, South Africa, despite ongoing national efforts. Based on a cross-sectional analysis of 415 mother–child pairs, the study found that 20–27% of children were malnourished, with 17% being severely affected. Low dietary diversity was observed in 82% of the sample. Malnutrition was associated with child age, maternal marital status, and maternal obesity. These findings emphasize the urgent need for integrated, household-level interventions to improve nutritional outcomes and infant and young child feeding (IYCF) practices. Therefore, the content of the study is appropriate for inclusion in this volume.

In the methodology section of the abstract, the authors indicate that a cross-sectional study was conducted on 415 mother–child pairs. However, it is important to specify the source population from which this sample was drawn, as well as the participation rate.

Response: the source population form which this sample was drawn from has been specified; Lines 106 – 108. Participation and recruitment rates have been mentioned in methods; lines 160 – 161.

The introduction appropriately outlines the significance of severe malnutrition among children under five in Africa, supported by relevant literature. However, given that the study concludes with a call for “integrated, household-level strategies to strengthen social protection and improve dietary diversity and IYCF practices,” the introduction should explicitly describe which such strategies were in place at the time of the study. This contextual information is necessary to assess the implications of the findings.

Response: We have added the strategies that were in place at the time of the study; lines 82 -89.

Furthermore, the final part of the introduction should clearly articulate the research hypothesis and the specific study objective. Currently, this section is confusing, and I recommend it be rewritten with particular attention to stating a clear and precise study objective.

Response; the research hypothesis and specified study objective have been added in lines 92 – 100.

In the Materials and Methods section, the study design is described as cross-sectional, conducted between 2022 and 2023. However, the manuscript does not describe the underlying population in the study area, which reportedly includes 18,797 children under the age of five. Ethical approval and informed consent procedures from parents or guardians should be explicitly documented.

Response: Underlying population in the study has been stated in lines 106-108. Although mentioned in lines 599 -608, we added ethical approval and informed consent in lines 108 – 114.

The sample size calculation assumes a 10% non-participation rate; however, the actual participation rate should be reported.

Response: Participation rate (100%) has been reported in line 161.

Similarly, in Figure 1, the origin of the source population should be clearly indicated.

Response: Source population added in figure 1, lines 170 - 171

In the data collection section, it is important to specify who collected the information and who performed anthropometric measurements on the children. The dietary questionnaire used was a 24-hour recall, which should be critically discussed, as this method is prone to bias and is generally considered less reliable than a seven-day recall.

Response: The principal investigator was responsible for measuring anthropometry for mothers and children (Lines 241-242 and 262 and 263). Research assistant conducted the interviews (Lines 211-213; 216 – 217). 24 hour recall information has been expanded (lines 225 – 231), and discussed in the limitations (lines 560 – 568)

While the procedures for measuring maternal and child anthropometry are described, it remains unclear who performed the measurements.

Response: The principal investigator was responsible for measuring anthropometry for mothers and children (Lines 241-242 and 262 and 263). Research assistant conducted the interviews (Lines 211-213; 216 – 217)

The statistical analysis section should detail the measures used to describe the results, particularly the prevalence. Since this is a cross-sectional design, prevalence ratios are more appropriate than odds ratios. Both crude and adjusted ratios should be presented to assess whether covariates modify the effect.

Response: Revised in lines 274 – 293. Crude and adjusted PR added in Table 7 and discussed; lines 432 – 454)

Tables 2 and 3 show the distribution of children by sex and nutritional status. These tables should include the criteria used to classify malnutrition, and this information should also be clearly stated in the Materials and Methods section. The p-value presented must indicate whether it refers to comparisons by sex or between the “yes” and “no” categories of each nutritional status group, to ensure the table is self-explanatory.

Response: added in material and methods – lines 255 – 258; implemented in Tables 2 (lines 334 – 338), and 3 (lines 356 – 361).

In Table 4, statistical tests should be used to compare the different variable categories presented.

Response: Implemented in Table 4 – line 380 - 385

In Table 7, both crude and adjusted risks should be reported, rather than only the adjusted ones.

Response: Implemented in lines 409 - 412

The discussion contextualizes the findings with relevant literature, acknowledging the limitations inherent to a cross-sectional design, which only allows for the identification of potential risk factors. However, the authors should be more cautious when drawing conclusions.

Response: we have revised the discussion to align with the results – lines 455 - 559

The conclusion should not merely summarize the study but rather provide a direct answer to the study objective.

Response: Revised in lines 578 - 591

In summary, the manuscript presents valuable insights into the nutritional status of mothers and children under five in a rural South African setting. However, several aspects require clarification and revision before the manuscript can be considered for publication.

Round 2

Reviewer 3 Report

Comments and Suggestions for Authors

I have carefully reviewed the revised version of the manuscript entitled “nutrients-3672362_ Acute Malnutrition in Under-Five Children in KwaZulu-Natal, South Africa: Risk Factors and Implications for Dietary Quality

”, as well as the authors' responses to the suggestions aimed at improving the clarity and understanding of the work presented.

Comments:

  • Regarding the use of the 24-hour dietary recall versus the 7-day recall questionnaire, the evaluation of their results is presented in the results section, whereas it would be more appropriate to include this discussion in the Discussion section. Furthermore, I do not agree with the assertion that the 7-day food variation assessment is inferior to the 24-hour recall. The 7-day recall is often conducted prospectively, and therefore, the authors should exercise caution in their interpretation.
  • In the tables, the last column should be labeled “p-value” rather than simply “p”.
  • The Discussion should take into account the nutritional intervention programs that were being implemented during the data collection period.
  • In the Conclusion, the contribution of the study’s findings to the existing body of knowledge should be clearly stated without reiterating the original objective. Additionally, the ongoing nutritional intervention programs during the data collection period should be considered in interpreting the results.

Author Response

  • Regarding the use of the 24-hour dietary recall versus the 7-day recall questionnaire, the evaluation of their results is presented in the results section, whereas it would be more appropriate to include this discussion in the Discussion section. Furthermore, I do not agree with the assertion that the 7-day food variation assessment is inferior to the 24-hour recall. The 7-day recall is often conducted prospectively, and therefore, the authors should exercise caution in their interpretation.

Response: We have revised the manuscript to relocate the discussion on the comparative strengths and limitations of the 24-hour dietary recall to the discussion section, where it is more appropriately contextualized. We have also revised the language to avoid implying that the 7-day recall is inherently inferior. Instead, we now acknowledge the methodological differences and contextual considerations that influenced our choice of the 24-hour recall method, while recognizing the strengths of the 7-day recall, particularly when conducted prospectively. Lines 502 – 506, and lines 583 – 588.

  • In the tables, the last column should be labeled “p-value” rather than simply “p”.

Response: We have revised the columns heading from “p” to “p-value” in the following tables to improve clarity and consistency:

Table 2: Comparison of nutritional indicators of children by sex

Table 3: Comparison of nutritional indicators of children by age

Table 4: Comparison of mothers’ sociodemographic characteristics

Table 5: Comparison of mothers’ obstetric history and selected feeding practices

Table 6: Comparison of means and proportions of DDS among children by sex and age

Table 7: Risk factors for acute malnutrition among children

  • The Discussion should take into account the nutritional intervention programs that were being implemented during the data collection period.

In response, we have revised the discussion section to include a paragraph that contextualizes our findings within the national nutrition interventions active during the study period (2022–2023), such as the Integrated Nutrition Programme (INP), the Child Support Grant (CSG), and the Community-based Management of Acute Malnutrition (CMAM). Lines 481 - 489

  • In the Conclusion, the contribution of the study’s findings to the existing body of knowledge should be clearly stated without reiterating the original objective. Additionally, the ongoing nutritional intervention programs during the data collection period should be considered in interpreting the results.

Response, we have revised the conclusion section to clearly articulate the study’s contribution to the existing body of knowledge, emphasizing its relevance to rural nutrition policy and practice. We have also incorporated a statement acknowledging the influence of ongoing national nutrition interventions during the data collection period. Lines 600 - 611

We sincerely thank you for your thoughtful and constructive comments during the first and second rounds of review. Your feedback not only strengthened the quality of our manuscript but also enhanced our understanding of key principles in conducting and presenting research. We genuinely appreciate your guidance and the opportunity to learn through this process.